# Advances in Noninvasive Carotid Wall Imaging with Ultrasound: A Narrative Review

**DOI:** 10.3390/jcm11206196

**Published:** 2022-10-20

**Authors:** Maria Alexandratou, Angeliki Papachristodoulou, Xin Li, Sasan Partovi, Andjoli Davidhi, Vasileios Rafailidis, Panos Prassopoulos, Vasileios Kamperidis, Ioanna Koutroulou, Georgios Tsivgoulis, Nikolaos Grigoriadis, Christos Krogias, Theodore Karapanayiotides

**Affiliations:** 1Department of Clinical Radiology, AHEPA University Hospital of Thessaloniki, Aristotle University of Thessaloniki, 541 24 Thessaloniki, Greece; 2Department of Radiology, Hospital of the University of Pennsylvania, Philadelphia, PA 19104, USA; 3Section of Interventional Radiology, Imaging Institute, Cleveland Clinic Main Campus, 9500 Euclid Avenue, Cleveland, OH 44195, USA; 41st Cardiology Department, School of Medicine, AHEPA University Hospital, Aristotle University of Thessaloniki, 546 36 Thessaloniki, Greece; 52nd Department of Neurology, School of Medicine, AHEPA University Hospital, Aristotle University of Thessaloniki, 546 36 Thessaloniki, Greece; 6Second Department of Neurology, School of Medicine, ‘Attikon’ University Hospital, National and Kapodistrian University of Athens, 157 72 Athens, Greece; 7Department of Neurology, St. Josef-Hospital Bochum, Ruhr University Bochum, 44801 Bochum, Germany

**Keywords:** carotid, atherosclerosis, ultrasound, contrast-enhanced ultrasound, stroke

## Abstract

Carotid atherosclerosis is a major cause for stroke, with significant associated disease burden morbidity and mortality in Western societies. Diagnosis, grading and follow-up of carotid atherosclerotic disease relies on imaging, specifically ultrasound (US) as the initial modality of choice. Traditionally, the degree of carotid lumen stenosis was considered the sole risk factor to predict brain ischemia. However, modern research has shown that a variety of other imaging biomarkers, such as plaque echogenicity, surface morphology, intraplaque neovascularization and vasa vasorum contribute to the risk for rupture of carotid atheromas with subsequent cerebrovascular events. Furthermore, the majority of embolic strokes of undetermined origin are probably arteriogenic and are associated with nonstenosing atheromas. Therefore, a state-of-the-art US scan of the carotid arteries should take advantage of recent technical developments and should provide detailed information about potential thrombogenic (/) and emboligenic arterial wall features. This manuscript reviews recent advances in ultrasonographic assessment of vulnerable carotid atherosclerotic plaques and highlights the fields of future development in multiparametric arterial wall imaging, in an attempt to convey the most important take-home messages for clinicians performing carotid ultrasound.

## 1. Introduction

Carotid atherosclerosis is an important cause of transient ischemic attack (TIA) and stroke, which are associated with significant morbidity and mortality. Albeit advances in cross-sectional imaging, ultrasound (US) remains at the forefront of screening, diagnosis, grading and follow-up of carotid atherosclerotic disease [1]. The technique’s value not only relies on cost-effectiveness, widespread availability, excellent safety profile and reproducibility but also on its evolving multiparametric nature. Multiparametric ultrasound combines anatomic information on B-mode and flow-visualization techniques with physiologic information acquired by pulsed wave Doppler techniques. Contrast-enhanced ultrasound (CEUS) and elastography represent recent advances adding to the spectrum of diagnostic information related to carotid atherosclerosis.

The term “vulnerable plaque” describes plaques associated with increased risk for brain ischemia. A number of imaging and histological features have been closely associated with increased stroke risk, dictating a paradigm shift from the traditional sole reliance on the degree of lumen stenosis. Such features include lipid content or intra-plaque hemorrhage, superficial ulcerations, fibrous cap thickness and intra-plaque neovascularization. Modern noninvasive US with its multiparametric capabilities is well-suited to provide accurate evaluation of the vulnerability of carotid plaques [1,2,3]. This comprehensive review presents current advances on vulnerable carotid atheroma detection focusing on plaque echogenicity, plaque surface morphology, CEUS and elastography; discusses the diagnostic value of different ultrasonographic techniques and illustrates characteristic clinical cases.

## 2. Plaque Echogenicity

Ultrasound remains the most commonly used imaging modality for the assessment of the extracranial carotid atherosclerotic disease, as it generates high-resolution images of atherosclerotic plaques providing information about their contour, composition and stenosis degree [4]. The percentage of luminal stenosis as a marker of carotid atherosclerotic burden has been criticized because both studies and clinical observations have shown that certain plaques producing milder degrees of stenosis may still lead to acute cerebral infarction [5]. Furthermore, the concept that embolic strokes of undetermined source (ESUS) are primarily of cardiogenic origin potentially meriting anticoagulation has not been confirmed by randomized clinical trials [6]. Conversely, nonstenosing thrombogenic atheromas may be the underlying pathology in a significant proportion of the etiologically heterogeneous ESUS population [7]. Large non-stenosing internal carotid artery plaques ipsilaterally to the side of cerebral ischemia have been identified in 35% of ESUS patients using CT angiography [8] and in 25% using color Doppler imaging [9]. Furthermore, intraplaque hemorrhage was identified using MRI in ipsilateral carotid atheromas in one out of five patients with ESUS [9]. Therefore, it is critical that besides luminal stenosis measurement, carotid plaque characteristics such as echogenicity, intraplaque neovascularization, ulceration and surface irregularity be evaluated and reported [5]. The association between imaging characteristics of atherosclerotic mural lesions and stroke risk to guide optimal management and prevention of associated brain ischemia is an evolving field of research [10]. The growth of atheromas leading to luminal stenosis and impairment of blood flow with altered hemodynamics is assessed by the well-established US velocity measurements [11]. Luminal stenosis caused by such atheromatous deposits can, within certain limits, be compensated by carotid wall remodeling processes including limited vessel wall expansion [11]. Nowadays, size of the plaque is no longer considered the main criterion for evaluating carotid disease and the risk for cerebrovascular ischemia [12]. A considerable percentage of stroke survivors with symptomatic carotid plaques have <70% stenosis [13]. Hence, traditional parameters used for the description of carotid atheromas (degree of stenosis, systolic peak velocity) seem to be insufficient predictors of the risk of embolization [13]. Rupture of the plaque can lead to thrombus formation, acute occlusion of the lumen (and/or) and ipsilateral embolic events [14]. The early detection and management of atheromas that are prone to rupture (“vulnerable”) may reduce the risk of future cerebrovascular events [14]. The main vulnerability features of carotid plaques are summarized in Figure 1.

Plaque formation is the result of a chronic progressive inflammatory process leading to deposits inside the sub-endothelial layer of the carotid wall consisting of lipids, connective tissue extracellular matrix (collagen, proteoglycans and fibronectin elastic fibers) and cells such as macrophages, T-lymphocytes and smooth muscle cells [15]. Plaque echogenicity is the imaging visualization of this process. The use of carotid plaque hypo-echogenicity as a risk stratification biomarker for predicting the annual risk for cerebrovascular ischemic events (stroke, TIA, amaurosis fugax) is supported by histopathologic studies showing that echolucent plaques mostly consist of lipid-rich necrotic cores (and/or) and intraplaque hemorrhage [15]. A meta-analysis involving 7557 asymptomatic patients followed for more than 3 years, demonstrated that plaques described as echolucent, showing intraplaque neovascularization and ulceration were associated with twice the risk of ischemic symptoms compared to stable echogenic plaques [16].

In a historical throwback toward the end of the 1980s, Gray-Weale et al. studied the importance of carotid plaque echogenicity showing a correlation between the preoperative ultrasound atherosclerotic carotid plaque appearance and carotid endarterectomy specimen histological characteristics [17]. They demonstrated that plaques of lower echogenicity were associated with an increased frequency of hemorrhage and lipid burden [17]. Based on B-Mode US, Gray-Weale and Nicolaides proposed a grading system based on echogenicity, that classified atherosclerotic plaques in five types: type 1 is uniformly echolucent; type 2 is predominantly echolucent with small areas of echogenicity; type 3 is predominantly echogenic with small areas of echolucency; type 4 is uniformly echogenic and type 5 consisted of plaques that could not be classified owing to heavy calcification and acoustic shadows [17,18] (Figure 2). Hypoechoic plaques (type 1 and 2) are associated with intraplaque hemorrhage and lipid accumulation, whereas hyperechoic homogeneous plaques are predominantly fibrous or calcified in nature [19]. As a result, the first two categories appear to be associated with a higher risk for surface disruption or rupture and yield a subsequent significantly higher risk of ipsilateral stroke when compared with non-echolucent plaques [17]. On the contrary, type 4 and 5 plaques are mainly encountered in patients with asymptomatic carotid disease [19,20]. Calcifications have been found to play an important role in plaque stabilization and lipid-rich plaques appear to be more often actively inflamed than either calcified or collagen-rich plaques (hyperechoic). Thus, heavily calcified carotid plaques could represent a chronic, less actively inflamed form of atherosclerosis [21]. Nevertheless, calcified intraluminal plaques may occasionally cause ischemia when the calcified material embolizes into the brain (Figure 3).

Echolucent unstable plaques with intraplaque hemorrhage and a lipid-rich necrotic core may cause microembolic phenomena to the arterial bed of the brain [22]. Patients with asymptomatic carotid plaques of low echogenicity have more frequently MRI (T2/FLAIR) T2 hyperintensities in the periventricular and subcortical white matter, silent lacunar lesions or cerebral microbleeds, conveying an increased risk of cognitive decline and vascular dementia [23]. Moreover, echolucent plaques are associated with an increased risk of stroke in patients undergoing carotid stenting and are associated with new cerebral ischemic lesions following endarterectomy [24]. A study performed on 1061 patients undergoing carotid endarterectomy associated plaque hypoechogenicity and ulcerations with the occurrence of new ischemic lesions on diffusion-weighted imaging 30 days post-surgery [25].

Another parameter widely used to assess carotid atheromas on B-mode ultrasound is the Gray Scale Median (GSM). It has been introduced to quantify plaque echogenicity in a more objective and reproducible manner. Quantitative assessment of the plaque is performed by a computer system assigning certain grayscale values to blood and adventitia [26]. GSM values from known tissue components are used and the measurement of the region of interest is expressed in a 256 gray-tone range where 0 is black and 255 is white. The GSM value of an entire plaque is obtained from a histogram calculated by software analysis. Plaques containing more calcium and fibrous tissue have higher GSM values, whereas plaques with richer lipid core and hemorrhagic components have lower GSM [26] (Figure 2). Atherosclerotic lesions with lower GSM are more prone to rupture and a lower GSM value may be considered an independent risk factor for stroke [27]. Carotid bifurcation plaques in patients with silent non-lacunar infarcts are usually hypoechoic (/) and of low GSM even in the absence of critical luminal stenosis [28].

During the last decade the term of juxtaluminal black area (JBA) has been introduced in the study of plaque echogenicity. It is defined as an area with a GSM value <25 adjacent to the lumen without a visible fibrous cap and has been linked linearly to elevated stroke risk [29]. Histologic studies performed on endarterectomy specimens have shown that JBA in ultrasound images is associated with lipid core proximity to the vascular lumen. The lipid rich necrotic core is closer to the lumen in symptomatic plaques causing thromboembolic phenomena in comparison to more stable asymptomatic plaques [30]. A study showed that the size of the JBA in asymptomatic carotid atheromas is linked to the possibility of a future ischemic event and can be used in stroke risk stratification models. A JBA > 4 mm is a considerable carotid disease indicator: the annual risk is 1.4% for patients with JBA of 4–8 mm, 3.2% for patients with JBA of 8–10 mm and 5% for patients with JBA > 10 mm [31].

The fibrous cap is a layer of fibrous connective tissue containing macrophages and smooth-muscle cells within a collagen-proteoglycan matrix associated with T-lymphocytes [30]. It covers the necrotic lipid core and constitutes a barrier separating the vascular lumen from the thrombogenic atheromatous contents of the plaque. Different caps vary in thickness, composition, and collagen content and thus in stability [30]. The rupture usually occurs in areas where the cap is the thinnest and often most heavily infiltrated by macrophage foam cells [31]. Fibrous cap thickness measurement of carotid atheromas with ultrasound is feasible, albeit technically demanding. Furthermore, discrimination of symptomatic from asymptomatic plaques based on ultrasound-measured mean cap thickness values is good and merits further development [32]. However, it must be noted that some fibrous caps may be so thin that they are usually not visible on classical ultrasound, while in heavily calcified plaques cap visualization may be impossible [33]. Newer high-resolution US devices with shear-wave elastography are able to visualize thick fibrous caps, especially in hypoechoic plaques (Figure 4).

A recently introduced imaging technology, MicroPure™ (Toshiba Medical Systems Corp., Tokyo, Japan) may improve visualization of microcalcifications on US [34]. This imaging technology allows the identification of the “Firefly sign”: microcalcifications are displayed as white dots in a blue background, similar to fireflies flickering in the dark. These signs are located in the fibrous caps of carotid atheromas and may be associated with plaque vulnerability. A 4-point Firefly score system has been developed and recent studies indicate that Firefly-positive atherosclerotic lesions are at an increased risk for rupture and embolic cerebral infracts [35].

Evaluating atherosclerotic carotid lesion echogenicity is not only useful as an indicator for its vulnerability and its association with ischemic events. A plaque that appears to become progressively more echogenic is possibly an indicator that its histological composition is changing and its stability is increasing [36]. Early and aggressive treatment with statins at high doses seems to increase the echogenicity of carotid plaques, making them less prone to rupture [36].

There has been a lot of research during the last years towards the use of radiomics and machine learning [37]. Carotid US being operator-dependent is expected to benefit from the use of artificial intelligence [38]. US-based radiomics models can be constructed by extracting features from grayscale images and may identify and quantify target features as the total plaque volume and composition (calcium, intraplaque hemorrhage, lipids) thereby predicting cerebrovascular ischemia risk [38]. Latest studies show that radiomics can reveal information invisible on advanced ultrasound imaging [37,38].

## 3. Surface Morphology

The histological definition of carotid plaque ulceration refers to the exposure of a plaque’s necrotic core to the circulation due to an endothelial defect of at least 1000μm in width [39,40]. This histologic entity is variably translated into various imaging criteria depending on the modality applied or the investigated study group [41,42]. In general, carotid plaques are characterized based on their surface morphology as smooth, irregular or ulcerated [43]. A smooth plaque exhibits a regularly outlined surface, whereas an irregular’s plaque outline fluctuates from 0.3 mm to 0.9 mm. An ulcerated plaque is one carrying a cavity measuring at least 1 or 2 mm, depending on the criteria followed [39]. The ulcer’s neck and base significantly vary in terms of shape and size, justifying the classification of ulcerations into distinct types [44] (Figure 5). Previously unclassified ulcerations such as “handle-shaped” can be occasionally seen, especially with the use of non-Doppler sensitive flow visualization techniques (Figure 6).

Carotid plaque ulceration has been long considered a major risk factor for stroke [45,46]. Carotid plaques with different ulcer depths have been correlated with embolic signals on transcranial Doppler US regardless of the degree of stenosis [47,48]. In a study where patients with symptomatic low-grade (<50%) carotid stenosis underwent endarterectomy, all plaques showed histological ulceration or rupture [49]. Of utmost importance is the (NASCET) The North American Symptomatic Carotid Endarterectomy Trial (NASCET) study, which established the higher risk for cerebrovascular events in patients with ulcerated plaques compared to those without ulceration, the risk further increasing in higher levels of stenosis [45,50]. A prospective multivariate analysis of patients with asymptomatic carotid stenosis significantly correlated ulceration with the onset of new neurologic symptoms [51]. Studies comparing symptomatic and asymptomatic carotid stenosis groups of patients showed higher frequency of ulcerated plaques in the former group, while ulcerations correlated with the occurrence of new events in the latter group [52,53]. Ulcerated plaques may actually increase the ipsilateral stroke risk up to seven times [54]. The combination of a hypoechoic plaque with an ulcerated surface on US appears to have a causative association with thromboembolic events, yielding a significant odds ratio of 9.34 [55]. In light of these results, carotid plaque ulceration should be sought with every available imaging modality, including US [1].

The role of mere irregularities without frank ulceration is controversial. Although initial studies proposed a degree of correlation with acute ischemic stroke, a later systemic meta-analysis defining the risk for stroke based on ultrasound characteristics of carotid atheromas concluded that only plaques with echolucency, intraplaque neovascularization, ulceration and intraplaque motion were associated with stroke [5]. One of the important studies to report that plaque irregularity on digital substraction angiography was associated with stroke was the ECST study [56].

US diagnostic accuracy for detecting carotid plaque ulceration is controversial, possibly due to the variety of techniques and diagnostic criteria used. It is undoubted though that the increased spatial and temporal resolution of modern ultrasound machines has improved image quality, owing to the wide use of higher-frequency transducers and advanced visualization technologies [21,39,57]. Initial publications showed that US is more accurate in diagnosing ulcers in plaques with <50% stenosis, though with poor correlation with histology, partially due to low intraobserver agreement for ulceration [58,59,60]. According to the criteria by De Bray et al. [42], an ulcer is a cavity > 2 mm in length and depth, with a well-defined back wall at its base on B-mode and flow reversal on Color Doppler technique. Although these criteria were widely adopted and used in the literature, they are associated with 35% sensitivity and 75% specificity [41,42]. Carotid atheroma ulceration criteria were renewed in 2012, achieving a sensitivity of 85% and specificity of 81%, despite the diagnostic limitation of acoustic shadowing caused by calcifications. The so-called Muraki criteria suggest that an ulcer is a cavity on plaque surface, regardless of its size, and that echogenicity at the cavity base should be less than that of the adjacent intima-blood border. The latter is characterized by higher acoustic impedance than that of ulcer basal thrombus or soft tissue. When the echogenicity criterion is not fulfilled there is the pitfall of characterizing as an ulcer what could be a mere atheroma indentation or two juxtapositioned but distinct lesions with normal endothelium between them (tandem plaques) [41]. As a consequence, an ulcer should only be diagnosed if it lies within the limits of a plaque and does not reach the level of intima-media interface and when it does form a cavity with sharp margins with or without overhanging edges. The term “yin-yang” sign is found in the literature describing this blood-flow reversal inside ulcerations, which can be demonstrated on most available flow visualization ultrasonographic techniques [2]. Another modern application of conventional US is the B-flow technique which demonstrates more accuracy in depicting plaque cavities owing to its improved flow sensitivity and both spatial and temporal resolution. More precisely, it depicts a swirling pattern of blood flow within the ulcer, in accordance with experimental and colour Doppler observations [61,62] (Figure 7).

Quantification analyses attempted to correlate atheroma surface morphology with symptomatology to improve inter-observer agreement. Tegos et al. used the approach of ‘bending energy’, which failed to achieve significant correlation with symptoms [63]. Conversely, Kanber et al. proposed the surface irregularity index (SII), a quantitative index taking into account the angular deviation of the plaque surface from a straight line, divided by the plaque’s surface length. SII provided an improved diagnostic accuracy for the detection of ipsilateral hemispheric cerebrovascular symptoms when combined with the degree of stenosis. Importantly, SII values were higher in symptomatic plaques but did not show association with the degree of stenosis, representing an independent risk factor [64]. Accordingly, SII was higher in symptomatic atheromas using both color Doppler and CEUS technique, whereas the subjective classification into smooth-irregular-ulcerated did not correlate with stroke occurrence [65]. Combining more than one indices into one composite index of “vulnerability” seems promising. Kanber et al. combined SII, degree of stenosis and GSM into a single index, which outperformed the degree of lumen stenosis alone for the detection of symptomatic plaques [66]. The same results were reproduced by a second study using both conventional color Doppler technique and CEUS. The latter achieved a slightly higher area under the curve, suggesting additional clinically significant information in the visualization of carotid plaque surface irregularities using CEUS compared with conventional techniques [67].

Last but not least, three-dimensional (3D) US is another promising modality in detecting ulcerated carotid plaques. 3D US is a new technique that multiplies the information taken from conventional 2D scanning. As a result, studies found that it reliably characterized plaque surface and defined ulceration in asymptomatic patients, offering slightly superior inter-observer reproducibility [68]. The addition of ulcer detection with 3D US appeared to increase the rate of patients with asymptomatic carotid stenosis who would benefit from interventional treatment [69].

## 4. Contrast-Enhanced Ultrasound (CEUS)

Intraplaque neovascularization and inflammation are the two important factors in plaque vulnerability. In comparison to normal vasa vasorum, aberrant micro-vessels are prone to hemorrhage, leading to increased risk of plaque rupture [70]. Inflammation can increase fibrous cap erosion and activate the platelet thrombogenic cascade, both important events in the pathogenesis of plaque rupture and thrombosis [71]. Detection and quantification of intraplaque neovascularization by CEUS may identify vulnerable atheromas. CEUS findings may correlate with pathology, intraplaque inflammation, and past cardiovascular or cerebrovascular events. Ongoing research focuses on the predictive value of CEUS concerning cardiovascular or cerebrovascular events.

### 4.1. CEUS and Histopathology

#### 4.1.1. Intraplaque Neovascularization

A recent meta-analysis has shown that the diagnostic odds ratio between intraplaque neovascularization and CEUS enhancement was 20.11 (95% confidence interval: 4.81–84.03) [72]. Further, studies have demonstrated direct correlation between the degree of CEUS intraplaque enhancement and the degree of histological intraplaque neovascularization, using semi-quantitative or quantitative measures. Semi-quantitative measurement typically utilizes visual grading of intraplaque enhancement. Shah et al. suggested a 4-tiered visual grading scale, with grade zero defined as no intraplaque enhancement, grade one defined as limited enhancement, grade two defined as moderate enhancement, and grade three defined as pulsating arterial enhancement [73]. The authors found a direct correlation between the degree of CEUS enhancement and histological micro-vessel density, measured by CD34 staining. Accordingly, another study found correlation between CEUS visual enhancement grading and histological micro-vessel density [74].

Quantitative measurement of CEUS enhancement is typically performed using specialized software, and varies widely. In one study, patient-specific parameters such as mean plaque filling, mean plaque intensity, and calcification percentage, were derived from time-intensity curves within intraplaque regions of interest. [75]. Another study found a strong correlation between quantitative CEUS enhancement and the degree of micro-vessel density, measured by CD34 staining, in consecutive patients undergoing evaluation for carotid endarterectomy [76,77].

#### 4.1.2. Intraplaque Inflammation

Carotid plaque inflammation documented by PET imaging is associated with increased risk of recurrent stroke at five years [78]. The association between CEUS enhancement and intraplaque inflammation is less clear. Using CD68 (a marker for the presence of intraplaque macrophage) as a surrogate for inflammation, a study found no correlation between CD68 and both semi-quantitative and quantitative CEUS intraplaque enhancement [74]. Similarly, Demeure et al. did not find correlation between CD68 staining intensity and qualitative CEUS intraplaque enhancement; however, they showed a direct correlation between CEUS intraplaque enhancement and histologic micro-vessel density [79]. Conversely, a study using the ratio of CEUS enhancement area to plaque area showed correlation between intraplaque enhancement and CD68 staining intensity [76]. Similarly, another study found correlation between late-phase CEUS intraplaque enhancement and markers of inflammation (CD68 and CD31 immuno-reactivity) [80]. In the future, targeted plasma proteomics in conjunction with ultrasound techniques may predict the development of carotid atheromatosis [81].

#### 4.1.3. Vulnerable Plaques

Instead of comparing CEUS intraplaque enhancement to markers of plaque vulnerability (neovascularization and inflammation), some studies examined directly the relationship between CEUS and histologically vulnerable plaque composition. However, the definition of histological vulnerability varied among studies and may have contributed to the heterogeneity in outcomes. A study defined vulnerability as a combination of large lipid core, increased inflammatory cell presence, and a lack of smooth muscle cells [82] whereas other studies used the American Heart Association (AHA) classification of atherosclerotic plaque [83,84].

Dynamic qualitative CEUS assessment had a sensitivity of 94.7%, a specificity of 76.9%, and an accuracy of 87.5% in diagnosing asymptomatic, histologically vulnerable plaques. Of note, on quantitative analysis, no significance was observed between histologically stable and unstable plaques [82]. In contrast, another study showed that CEUS intraplaque enhancement was associated with histological vessel density in AHA grade V (non-vulnerable) but not in AHA grade VI (vulnerable) plaques [85]. D’Oria et al. studied consecutive asymptomatic carotid stenosis patients who underwent carotid endarterectomy. Qualitative CEUS intraplaque enhancement was compared to histological analysis (AHA classification of atherosclerotic plaques). The authors showed that vulnerable plaques (AHA grade VI) were associated with increased micro-vessel density compared with those with non-vulnerable plaques (AHA grade IV and V) (*p* = 0.004). Nevertheless, there was no significant difference in CEUS enhancement between patients with vulnerable and non-vulnerable plaques [84].

#### 4.1.4. Plaque Ulceration

Many recent studies have investigated the role of CEUS in detecting carotid plaque ulceration and simple surface irregularities [86] (Figure 8). Hamada et al. compared histological results to US and CEUS, confirming CEUS superiority and calculating optimal cut-off values of a cavity’s orifice (1.4 mm), depth (1.3 mm) and width (1.88 mm). If one of these cut-off values is exceeded, the sensitivity of CEUS for diagnosis of fibrous cap disruption is 91% [87]. Other studies used multi-detector CT angiography (MDCTA) as the reference method and confirmed that CEUS improved diagnostic accuracy for the diagnosis of plaque ulceration [2,88,89]. The widely accepted definition for ulceration using CEUS is the projection of microbubbles columns within the plaque measuring at least 1 × 1 mm. [29,31] CEUS can also depict a swirling pattern of microbubbles in 18% of ulcer-associated cavities, indicative of the underlying mechanism of arterio-arterial embolization [90,91]. Τhe sensitivity of CEUS for ulcer detection was 88% in a symptomatic patient population compared to 29% for color Doppler [88], and 94.1% in a mixed symptomatic and asymptomatic patient population compared to 41% for color Doppler [91]. CEUS offers excellent spatial and temporal resolution, achieving nearly real-time imaging, within the focused field-of-view containing a carotid plaque. A recent study showed that the direction of microbubble flow is important and is associated with the likelihood for plaque rupture and with vulnerability features on histology. Microbubbles flowing from the lumen toward the center of the plaque are associated with fibrous cap rupture with a sensitivity of 87.5% and a specificity of 92.6%. The “inside-out” microbubble direction pattern is an independent risk factor for plaque rupture, yielding an OR of 8.5 [92]. Modern technology can create parametric color maps objectively visualizing the flow pattern of microbubbles (Figure 9).

### 4.2. CEUS and Intraplaque Inflammation

Though CEUS is a strong surrogate of intraplaque neovascularization, this is likely less so for intraplaque inflammation. Multiple CEUS parameters, such as late-phase enhancement may serve as markers of plaque inflammation [93]. Pathogenesis of vulnerable atheromas is an intricate interplay between carotid neovascularization and local inflammation. There may be a temporal difference between the former and the latter implying that the presence of neovascularization does not necessarily predispose to plaque rupture if the concurrent localized inflammation is minimal; alternatively, neovascularization could be sequelae of prior inflammation. The above may explain why up to 40% of asymptomatic plaques are associated with intraplaque neovascularization [94]. It is now clear that there is an association between inflammatory markers in atherosclerotic patients and stroke occurrence rate [95]. Furthermore, it is impressive that circulating proteins have been found able to predict the development of preclinical atherosclerosis [96].

#### 4.2.1. Serum Inflammatory Markers

Systemic serum inflammatory markers are possible surrogates for intraplaque inflammation. One of the best studied serum inflammatory markers is C-reactive protein (CRP) and elevated levels have been associated with plaque vulnerability [96]. Chang et al. have shown significant correlation between semi-quantitative CEUS intraplaque enhancement and serum CRP levels [97]. Similarly, another study examined the relationship between high-sensitivity CRP and qualitative or quantitative CEUS findings in patients with and without acute ischemic stroke. The authors found significant differences in serum CRP levels between enhancing and non-enhancing plaques in patients with acute ischemic stroke, Furthermore, the peak intensity ratio was found to correlate with serum CRP levels. Of note, this relationship was not significant in the control group [98]. Other inflammatory markers have been studied as well. A recent study showed significant correlation between quantitative CEUS intraplaque enhancement and circulating lymphocyte and neutrophil counts [99].

#### 4.2.2. PET Imaging

Positron-emission tomography (PET) imaging of the carotid artery is capable of detecting localized intraplaque inflammation and may enable risk stratification, albeit at a significant cost [100]. Comparison of nuclear medicine and CEUS studies yielded conflicting results. A small study of 13 patients showed positive correlation between quantitative CEUS enhancement and the degree of 18-fluorofeoxyglucose (FDG) uptake in the carotid plaque [101]. Conversely, a more recent study of 30 patients utilizing multimodality imaging and histological analysis of carotid plaques found that the mean FDG uptake was similar between patients with CEUS intraplaque enhancement and those without. The study documented association of CD68 immuno-positivity with FDG uptake but not with CEUS intraplaque enhancement [79].

### 4.3. CEUS and Clinical Events

#### 4.3.1. Prior Cardiovascular/Cerebrovascular Events

One of the earliest studies which evaluated the potential association between qualitative CEUS enhancement and history of cardiovascular events (CVE: myocardial infarction, stroke or transient ischemic attack) found that patients with intraplaque enhancement had higher incidence of prior CVE than those without. However, there was no significant association between intraplaque enhancement and history of cardiovascular disease (CVD: CVE, coronary artery disease, peripheral arterial disease) [102]. Another study found that patients with symptomatic carotid plaques had higher quantitative CEUS intraplaque enhancement compared to those with asymptomatic plaques [77]. Symptomatic plaques were defined as those with major or minor stroke, TIA, or amaurosis fugax within the past 6 months. Conversely, Demeure et al. showed no correlation between qualitative CEUS intraplaque enhancement and past cardio/cerebrovascular events in a small number of patients [79]. Nevertheless, a recent meta-analysis showed that the presence of intraplaque enhancement was associated with previous cardiovascular (OR: 4.25, 95% CI: 2.48–7.29) and cerebrovascular events (OR: 4.83, 95% CI: 2.66–8.78) [103].

#### 4.3.2. Future Cardiovascular/Cerebrovascular Events

An early study on the predictive value of CEUS on cardiovascular events enrolled a total of 304 patients. The authors found significant correlation between the degree of CEUS intraplaque enhancement and coronary extent score, number of complex coronary lesions, and number of diseased coronary arteries. Importantly, 84 patients experienced acute coronary syndrome (ACS) during the follow-up period. Multivariate analysis showed that higher grade of CEUS intraplaque enhancement was an independent risk factor for ACS (OR: 1.91, 95% CI: 1.04–3.53) [104]. A recent larger study by Mantella et al. confirmed that increased CEUS intraplaque enhancement was associated with significant (>50% stenosis) coronary artery disease (CAD). Kaplan-Meier analysis showed that patients with CEUS intraplaque enhancement score > 1.25 had higher risk of CAD with a sensitivity of 92% and a specificity of 89% [105]. Similarly, a recent study showed that increased CEUS intraplaque enhancement was associated with higher risk of CAD during follow up (OR 4.88, 95% CI: 1.77–13.49) [106]. Regarding cerebrovascular events, Camps-Renom et al. showed that CEUS intraplaque neovascularization was an independent predictor for recurrent stroke (hazard ratio, 6.57, 95% CI: 1.66–26.01) [107]. Similarly, a more recent study found that grade 2 intraplaque enhancement (extensive enhancement) was associated with ischemic stroke recurrence (hazard ratio 4.54, 95% CI: 1.89–10.87) [108].

## 5. Elastography of Carotid Atherosclerotic Disease

Elastography can be used to assess the stiffness of a plaque, reflective of its histologic composition. Elastography evaluates mechanical properties by measuring plaque displacement and deformation. Two elastographic methods exist for the measurement of elastic deformation of a tissue: strain (SE) and shear wave elastography (SWE) [109,110]. SE measures plaque displacement caused by an external force, such as blood pressure oscillations or manual compression of the probe. It –semi-quantitative parameters such as strain, strain velocity or strain rate through deformation estimating algorithms. In SWE, the transducer emits shear waves through an acoustic radiation force impulse [111]. These waves disseminate perpendicularly to the impulse. The technique measures the velocity with which shear waves propagate through the tissue, expressed as Young’s modulus (YM) [109]. YM defines the tissue resistance to elastic deformation: it quantifies the amount of stress needed to achieve a unit of deformation, essentially measuring tissue elasticity. Soft tissues, such as plaque lipid core, demonstrate significant elastic deformation, lower YM and lower shear wave velocities (SWV), whereas more rigid tissues such as calcified plaques demonstrate less elastic deformation and higher SWV [112] (Figure 10).

Ultrasound elastography can assess carotid atherosclerotic plaques in combination with other techniques in the setting of multi-parametric ultrasound [113,114]. However, sensitivity and specificity vary: using MRI as the reference, sensitivity was 71.4% and specificity was 87.1%; using histology, sensitivity decreased to 50% while specificity reached 100% [115,116]. Significantly lower mean YM and SWV values were found in symptomatic plaques compared to the asymptomatic group [117,118,119,120]. Shang et al. demonstrated lower SWV values in hypoechoic plaques suggesting that SWE indices could be used to discriminate vulnerable from less vulnerable plaques. The same study compared SWV values to homocysteine serum levels and found a negative correlation with higher homocysteine values associated with lower SWV values in carotid plaques of stroke patients [118]. Ramnarine et al. demonstrated the value of SWE imaging to identify carotid plaques prone to rupture by correlating YM values with the Gray-Weale echogenicity grading and GSM values [119]. Another study showed that YM was superior vulnerability marker than GSM, and that combining YM values with the degree of stenosis improved diagnostic performance [118]. Doherty et al. showed increased SWE displacements in regions identified as lipid on MRI, while Huang et al. suggested that larger local deformations and increased complexity in deformation patterns are more likely to occur in vulnerable plaques [121,122,123]. Using histology as the reference method, Czernuszewicz et al. showed increased diagnostic accuracy of elastography when fibrous cap thickness was included in the measurements, with smaller thickness associated with higher rupture risk at a cut-off value of 0.5 mm [124]

Although quantification of elastography indices will not replace grading of stenosis to determine eligibility for surgery, the additional information provided may improve detection of vulnerable plaques and patient risk stratification. To date, however, most studies include small number of patients limiting the clinical value of elastography and warranting larger studies with longitudinal follow-up.

## 6. Conclusions

The entity of vulnerable carotid plaque is currently well-established and the associated cerebrovascular risk assessment could rely on multiparametric ultrasound examination. Recent developments have highlighted a spectrum of ultrasound characteristics contributing to plaque risk for rupture and extending far beyond the degree of lumen stenosis. Further research will consolidate their clinical value and steer the incorporation of novel ultrasound modalities into daily practice assessment of carotid atherosclerotic disease.

## Figures and Tables

**Figure 1 jcm-11-06196-f001:**
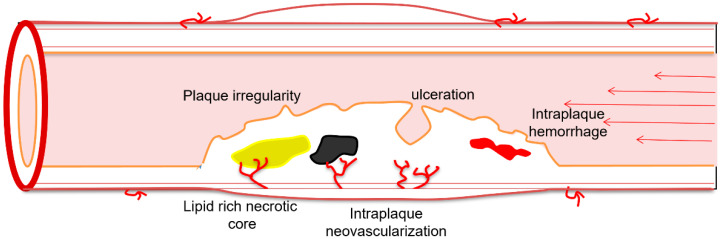
Schematic drawing showing the vulnerability features of carotid plaque: intraplaque hemorrhage appears as a red area; lipid rich necrotic core appears as a yellow area; plaque surface irregularity and ulceration appears as a line covering the plaque; intraplaque neovascularization appears as thin red branching vessels inside the plaque. Calcification with acoustic shadowing is also included but is a protective feature of carotid atherosclerosis.

**Figure 2 jcm-11-06196-f002:**
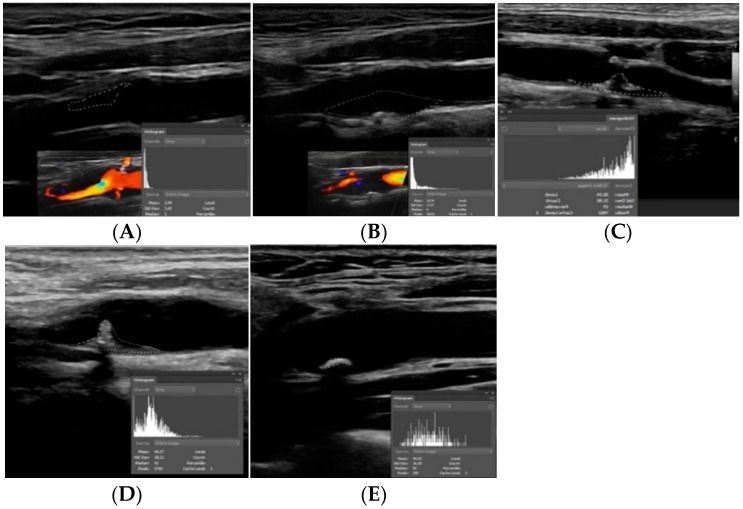
Subjective and objective assessment of carotid plaque echogenicity. Type 1 (**A**) refers to a uniformly echolucent plaque. Type 2 (**B**) is a plaque mainly echolucent with small areas of echogenicity. Type 3 (**C**) is predominantly echogenic with small areas of echolucency. Type 4 (**D**) is a uniformly echogenic plaque. Type 5 (**E**) is a plaque that cannot be otherwise classified owing to heavy calcification and acoustic shadow. Images presented are B-mode scans. In A and B colour Doppler was also used to delineate the plaques. Embedded panels at the right down corner represent Gray Scale Median (GSM) histograms. Notice that in type 1 and 2 plaques the histogram is heavily skewed to the left (lower GSM values, closer to total black) whereas in type 3 and 4 plaques GSM values are more evenly distributed or even skewed to the right (higher GSM values, closer to total white).

**Figure 3 jcm-11-06196-f003:**
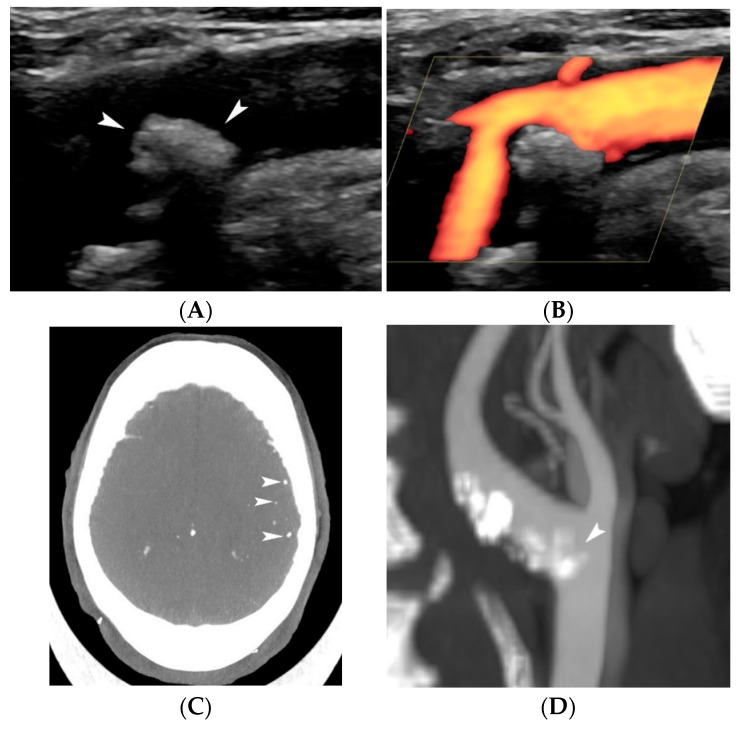
A free-floating calcified plaque causing embolic stroke. B-mode image (**A**) showing a free-floating intraluminal calcified plaque in the origin of the internal carotid plaque (arrowheads). Power Doppler technique (**B**) showing the moderate stenosis caused. Brain Computed Tomography (**C**) performed in the setting of multiple acute left-sided strokes showing multiple calcific emboli (arrowheads) not seen in previous scans. Computed Tomographic Angiography (**D**) confirming the intraluminally projecting calcified plaque (arrowhead) at the origin of the internal carotid artery.

**Figure 4 jcm-11-06196-f004:**
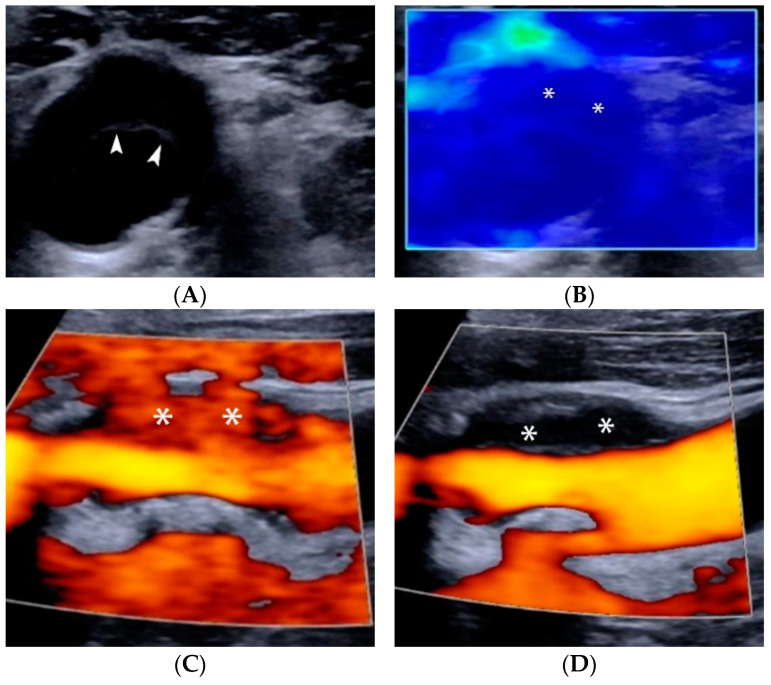
Ultrasonographic assessment of plaque stiffness. B-mode axial image (**A**) showing a hypoechoic plaque causing positive remodeling. Note a thin echogenic layer representing the plaque’s fibrous cap (arrowheads). Shear-wave elastography (**B**) showing low shear-wave velocity values and thus lower stiffness in the plaque’s core (dark blue colour, asterisks) but a slightly higher value and stiffness for the fibrous cap (slightly brighter blue layer covering the core on the luminal aspect of the plaque). The Power Doppler vocal fremitus artifact could be appreciated with the soft plaque covered by signals when the patient spoke (**C**, asterisks). A smooth surface of the plaque (asterisks) was appreciated when the patient was still (**D**).

**Figure 5 jcm-11-06196-f005:**
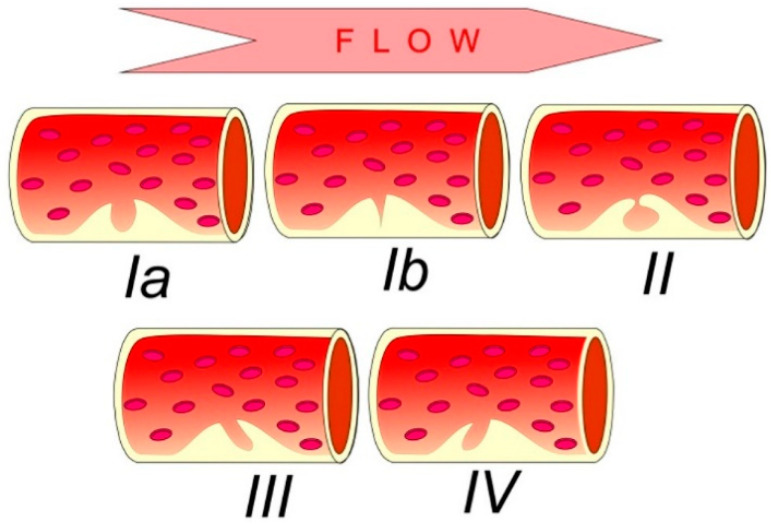
Schematic diagram showing the five morphologic types of carotid plaque ulcerations. Ulcer perpendicular to lumen with parallel (type Ia) or converging (type Ib) sides. Type 2: ulcer with narrow neck (“mushroom shaped”). Ulcer oriented parallelly (type III) or antiparallelly (type IV) to blood flow direction. Familiarity with these types will help identification of ulcerations.

**Figure 6 jcm-11-06196-f006:**
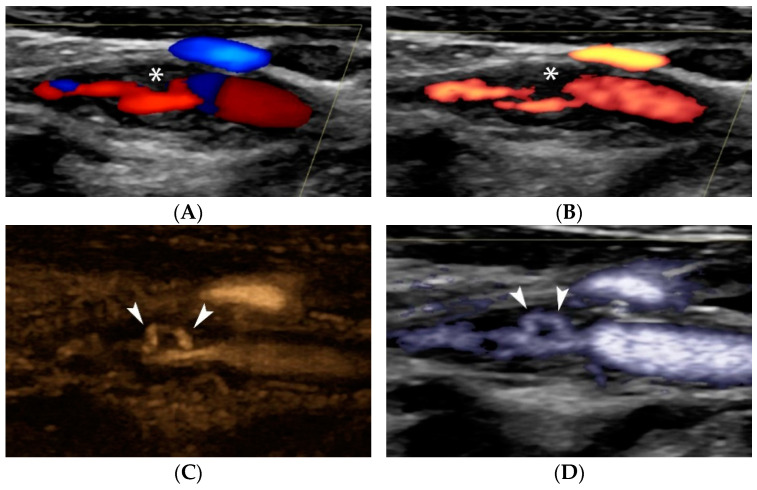
“Handle-shaped” ulceration visualization on Doppler and non-Doppler techniques. Colour Doppler image (**A**) showing a moderately stenotic internal carotid artery plaque with irregular surface (asterisk). Power Doppler technique (**B**) confirming the findings but visualizing more pronounced irregularity (asterisk). B-Flow technique with (**C**) and without static tissue suppression (**D**) show that there is actually a tunnel-like ulceration (arrowheads), also partially identified on CTA (**E**) as an area of enhancement (arrowhead) inside the plaque. Note the blurred appearance of ulceration on CTA due to the lower spatial resolution of the technique as compared with ultrasound, which offers excellent spatial resolution in the field-of-view, particular in the near field.

**Figure 7 jcm-11-06196-f007:**
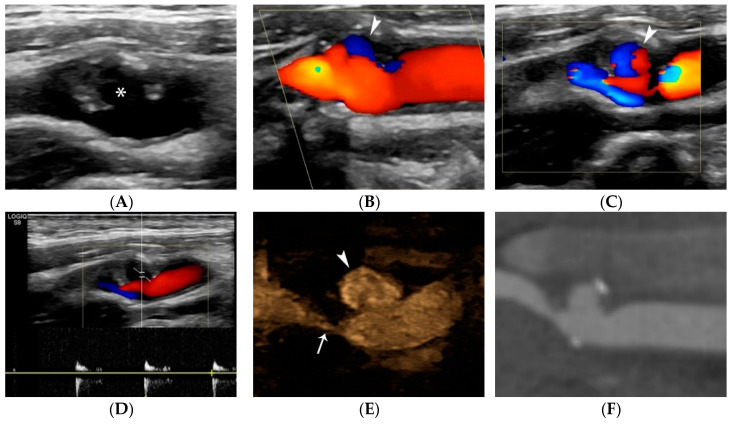
A 57-year-old man undergoing pre-operative carotid ultrasound. B-mode image (**A**) showing a severely stenotic internal carotid artery plaque with a large ulceration cavity (asterisk). Colour-Doppler technique images showing the classic ulceration signs of blood flow reversal (**B**) (arrowhead) and “yin-yang” sign (**C**) (arrowhead). Pulsed wave Doppler technique (**D**) showing the “to-and-fro” flow pattern. B-Flow technique (**E**) accurately visualizing full extent of the ulceration (arrowhead) and stenotic lumen (arrow), without overwriting, aliasing or Doppler-Dependence artifact. The need for echocontrast agent (microbubbles) is obviated. CTA image (**F**) confirming the findings and closely correlating with US.

**Figure 8 jcm-11-06196-f008:**
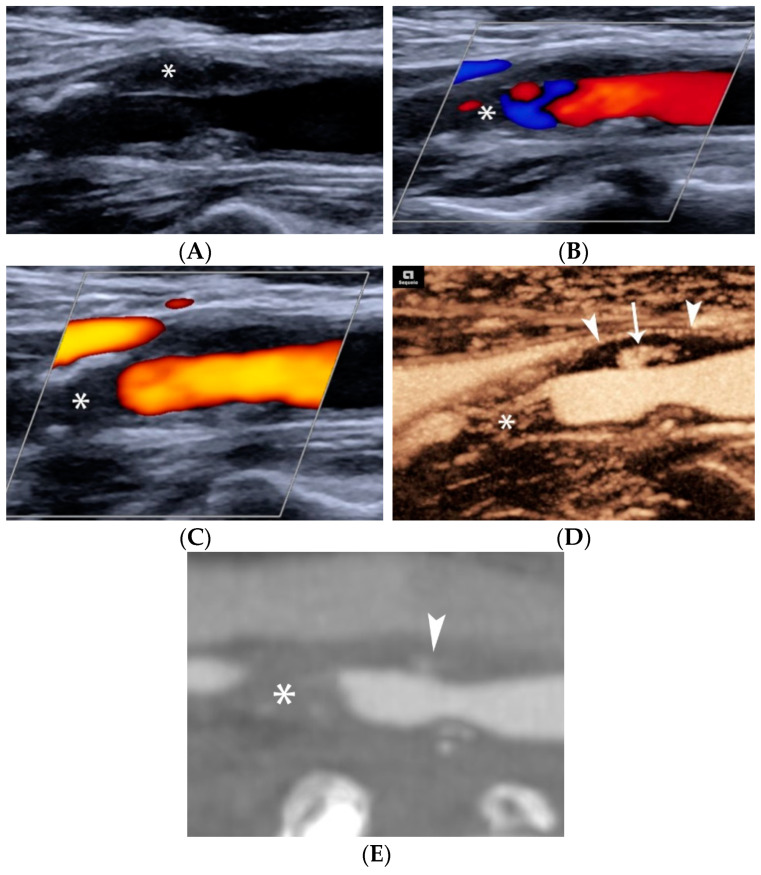
A 60-year-old man with stroke. B-mode (**A**) showing an elongated hypoechoic plaque in the near carotid wall (arrowheads). Note the echogenic thin fibrous cap and the distal calcification (asterisk). Colour (**B**) and power Doppler (**C**) images showing a smooth surface with limited distal flow signals, raising suspicion of occlusion (asterisk). Maximum-Intensity-Projection (MIP) CEUS image (**D**) showing a pre-occlusive stenosis (asterisk) with patent distal lumen, along with a small superficial ulceration (arrow) and adventitial neovascularization (arrowheads). CTA (**E**) confirming the presence of ulceration (arrowhead) and pre-occlusive stenosis (asterisk). Note the blurred appearance of ulcer outline owing to the lower spatial resolution of CTA compared to CEUS.

**Figure 9 jcm-11-06196-f009:**
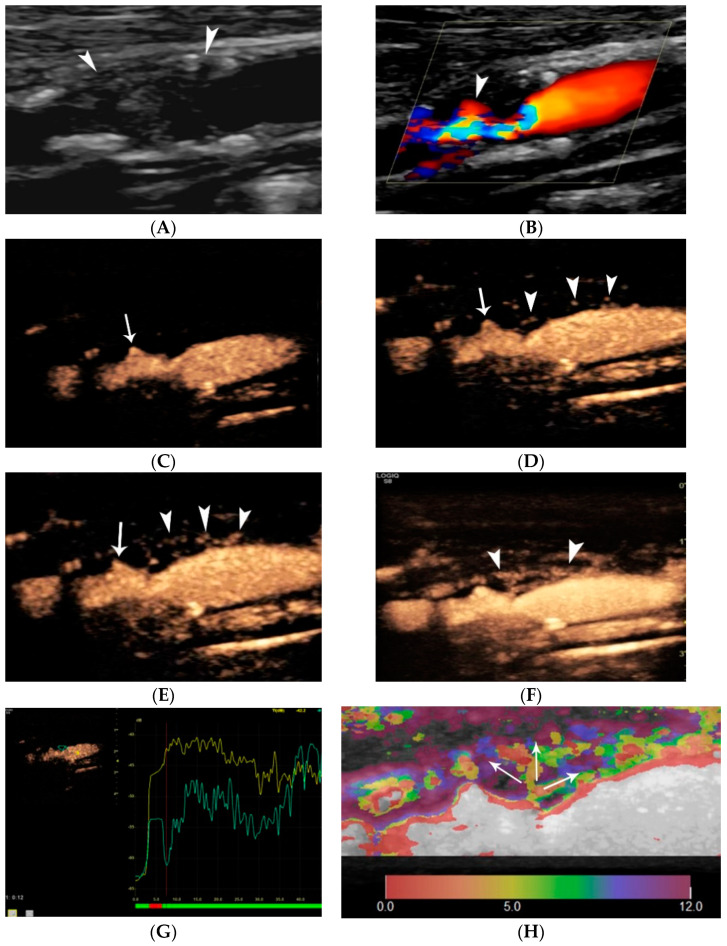
A 65-year-old man with stroke. B-mode US (**A**) shows an internal carotid plaque of mixed echogenicity causing severe stenosis. Colour-Doppler technique (**B**) shows blood flow turbulence confirming haemodynamically significant stenosis. Note the presence of a superficial ulceration (arrowhead). Consecutive CEUS images (**C**–**E**) confirm the presence of an ulceration (arrow), while microbubbles are progressively seen plaque core and adventitial layer, in keeping with severe intra-plaque neovascularization (arrowheads). Temporal Maximum Intensity Projection Image (**F**) showing all the microbubbles visualized within a specific time frame and thus creating a vascular map of intra-plaque neovascularization (arrowheads). Time-intensity Curve analysis (**G**) showing the enhancement of the plaque (green line) compared with the lumen (yellow line), providing quantitative data. Colour-coded parametric image (**H**) showing the flowing pattern and direction of microbubbles inside the plaque core based on their time of arrival. A direction from lumen towards the adventitia can be appreciated.

**Figure 10 jcm-11-06196-f010:**
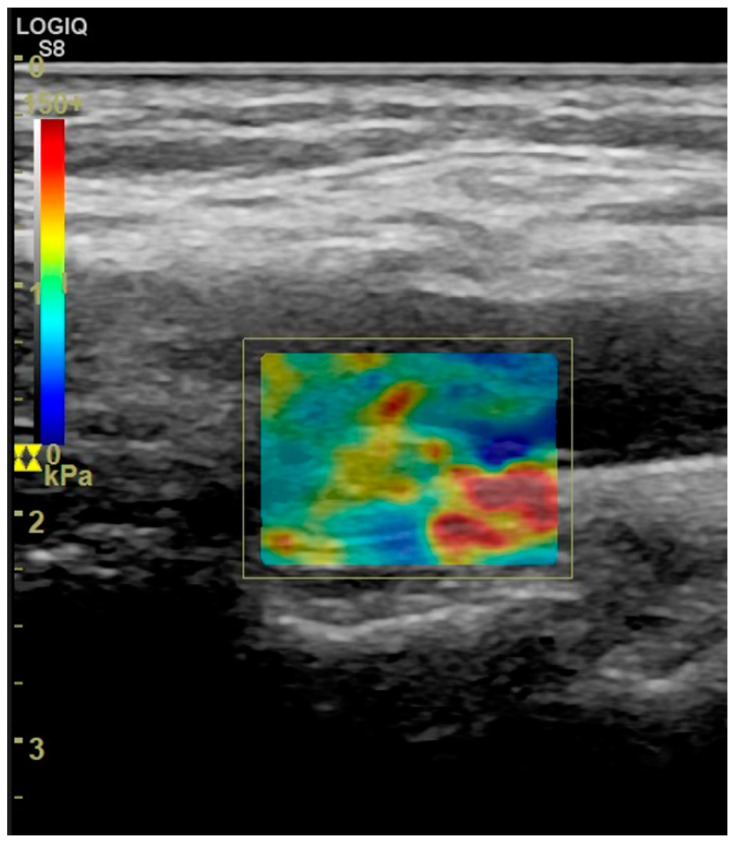
Shear-wave elastography image showing an echogenic plaque. The plaque yields high shear wave velocity values (yellow to red colour in left-hand scale), suggesting that this is a stiff plaque containing fibrous and calcific tissue.

## Data Availability

Not applicable.

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
