# Peer review of "Advances in Noninvasive Carotid Wall Imaging with Ultrasound: A Narrative Review"

_jcm, 2022, doi:10.3390/jcm11206196_

Round 1

Reviewer 1 Report (Previous Reviewer 3)

I have no further comments regarding the revised manuscript.

Reviewer 2 Report (Previous Reviewer 2)

Authors addressed all my previous comments 

I have no further comment to do 

This manuscript is a resubmission of an earlier submission. The following is a list of the peer review reports and author responses from that submission.

Round 1

Reviewer 1 Report

In the attached file, all the comments are given.

Reviewer 2 Report

Authors performed a  comprehensive review  of carotid wall imaging with ultrasound.

This is a well performed and really complete review and I have only minor comment to do: 

1) authors should add with regard of ESUS strokes some further few sentences about the real burden of cardioembolic pathogenesis over atherothromboembolic pathogenesis in this stroke subype with a pathogenesis not yet full known.

2) authors should add a sentence about the role of inflammation markers in atherosclerotic patients with regard stroke rates and they should add these citations on their reference section :

Siragusa S, Malato A, Saccullo G, Iorio A, Di Ianni M, Caracciolo C, Coco LL, Raso S, Santoro M, Guarneri FP, Tuttolomondo A, Pinto A, Pepe I, Casuccio A, Abbadessa V, Licata G, Battista Rini G, Mariani G, Di Fede G. Residual vein thrombosis for assessing duration of anticoagulation after unprovoked deep vein thrombosis of the lower limbs: the extended DACUS study. Am J Hematol. 2011 Nov;86(11):914-7.; 

3) Authors should add a sentence about the predictive role of inflammatory proteomic of the carotid plaque toward stroke incidence and recurrence on the basis of this very interesting and recent article published by Stroke in 2022 (Baragetti A, Mattavelli E, Grigore L, Pellegatta F, Magni P, Catapano AL. Targeted Plasma Proteomics to Predict the Development of Carotid Plaques. Stroke. 2022 Jul 1:101161STROKEAHA122038887)

Reviewer 3 Report

This review is a useful summary of recent findings on carotid echocardiography, but it is too broad and difficult to understand in some areas.

#1 The main sections are subtitled: plaque echogenicity, surface morphology, contrast-enhanced ultrasonography (CEUS), and elastography, but there are many references to CEUS. CEUS is discussed extensively. I think CEUS is also mentioned in relation to inflammatory markers and cardiovascular events. Inflammation of the carotid artery seems to be involved in plaque echogenicity, surface morphology, CEUS, and Elastography. There seems to be no need to address it in CEUS. The authors should change to an appropriate structure.

#2 In surface morphology and CEUS, detection of ulceration seems to be significant. I would like to see data on the detection of ulceration in comparison with MRI and contrast-enhanced CT and on the prediction of cerebral infarction based on plaque echogenicity compared with MRI and contrast-enhanced CT. If you have such data, it would be easier to understand if you could show them in a table.

#3 In Figure 1, the Gray Scale Median histogram seems small. Please trim or put it on the side to make it easier to see.

#4 It is difficult to understand B in Figure 3. Please enlarge it or add an explanation.

#5 It is written as a "firefly sign" on p.7, but I think it would be easier to understand if you could illustrate it.

#6 Elastography is also easy to understand if there is some kind of figure.